# Catalytic Pyrolysis of PET Polymer Using Nonisothermal Thermogravimetric Analysis Data: Kinetics and Artificial Neural Networks Studies

**DOI:** 10.3390/polym15010070

**Published:** 2022-12-24

**Authors:** Ibrahim Dubdub, Zaid Alhulaybi

**Affiliations:** Department of Chemical Engineering, King Faisal University, Al-Ahsa 31982, Saudi Arabia

**Keywords:** PET, catalytic pyrolysis, zeolite β, kinetics, TGA, ANN

## Abstract

This paper presents the catalytic pyrolysis of a constant-composition mixture of zeolite β and polyethylene terephthalate (PET) polymer by thermogravimetric analysis (TGA) at different heating rates (2, 5, 10, and 20 K/min). The thermograms showed only one main reaction and shifted to higher temperatures with increasing heating rate. In addition, at constant heating rate, they moved to lower temperatures of pure PET pyrolysis when a catalyst was added. Four isoconversional models, namely, Kissinger–Akahira–Sunose (KAS), Friedman, Flynn–Wall–Qzawa (FWO), and Starink, were applied to obtain the activation energy (*E_a_*). Values of *E_a_* acquired by these models were very close to each other with average value of *E_a_* = 154.0 kJ/mol, which was much lower than that for pure PET pyrolysis. The Coats–Redfern and Criado methods were employed to set the most convenient solid-state reaction mechanism. These methods revealed that the experimental data matched those obtained by different mechanisms depending on the heating rate. Values of *E_a_* obtained by these two models were within the average values of 157 kJ/mol. An artificial neural network (ANN) was utilized to predict the remaining weight fraction using two input variables (temperature and heating rate). The results proved that ANN could predict the experimental value very efficiently (R^2^ > 0.999) even with new data.

## 1. Introduction

Plastic waste has become a serious global problem, especially in countries where a huge quantity of plastic waste is produced and disposed. There are many academic research centers attempting to find a suitable way to recycle such vast waste [1].

PET is one of six polymers, with the others being low-density polyethylene (LDPE), high-density polyethylene (HDPE), polystyrene (PS), polypropylene (PP), and polyvinyl chloride (PVC). PET accounts for almost 7.6 wt% of plastic solid waste (PSW) [2,3]. It is very commonly used in manufacturing food packaging and fiber cloth and is considered as the most recycled material [4]. PET is mostly used in manufacturing fiber, staple filaments, textile, and carpet [3,5].

TGA is a thermal analysis method that is best known for calculating the kinetic parameters of plastic pyrolysis. Estimation of kinetic parameters from TGA data can be performed by different methods. These methods use either multiple TGA datasets at different heating rates (called “model-free” or “isoconversional” method) or one single TGA dataset (called “model-fitting” method) and generate the final kinetic equation using either differential method or integral technique.

In the following literature survey, only the kinetics of PET by TGA data and the kinetics of catalytic cracking of different plastic waste polymers by TGA data are examined. Yang et al. (2001) [6] initiated the kinetic study of PET using the Arrhenius equation with Friedman’s method for DTG curves to obtain the kinetic parameters of PET polymers. TG and DTG curves showed only one single decomposition that started at 370 °C, reached maximum at 441 °C, and ended at 502 °C at 10 °C/min heating rate. The authors obtained activation energy value of 242 kJ/mol with reaction order (n) of 1. Martín-Gullón et al. (2001) [2] showed that the pyrolysis reaction could almost be divided into two reactions. The first one with 80% of weight loss, considered as the main reaction within the range of the temperature (700–750 K), had activation energy of 256.4 kJ/mol, while the second one with only 6% weight loss had activation energy of 332.3 kJ/mol. Girija et al. (2005) [7] highlighted that PET had only single-stage degradation at temperature of 440 °C with weight loss of 60%. Saha and Ghoshal (2005) [8] studied the pyrolysis kinetics for different soft drink bottles to obtain the activation energy value using different models. They reported *E_a_* within the values of 320–339.89 kJ/mol.

Aguado et al. (2007) [9] measured the activity of two catalysts, namely, n-HZMS-5 and Al-MCM-41, on the degradation of LDPE. This activity was measured by how much the TG shifted to lower temperature. They stated that the pure polymer started to degrade at 477 °C and shifted to 419 °C with Al-MCM-41 and then further to 396 °C with zeolite n-HZSM-5. They attributed this strong activity for zeolite n-HZSM-5, which has weak acid properties, to the large pore dimension. Moltó et al. (2007) [10] investigated the kinetics of PET fiber cloth pyrolysis with three different heating rates. They found three different partial weight loss reactions, with the second reaction being the main one within the range of 640–730 K temperature. They first suggested three independent reactions with activation energy values of 121.5, 238.3, and 300.0 kJ/mol but later considered the second reaction with 238.3 kJ/mol as the main reaction. These results were close to the findings of Yang et al. (2001) [6], who showed an energy value of 242 kJ/mol.

Halász et al. (2009) [11] repeated the same experiment carried out by Aguado et al. (2007) [9] but with PET polymer. They found that pure polymer PET had DTG peak maximum equal to 473 °C, which shifted to 412 °C with H-ZSM-5 and to 458 °C with Ti-MCM-41.

Çepelioğullar and Pütün (2013) [12] investigated the effect of different biomasses (cotton stalk (CS), hazelnut shell (HS), sunflower residue (SFR), and arid land plant *Euphorbia rigida* (ER)) in the copyrolysis with PET. They mentioned that the pyrolysis of PET decomposed at two stages, which started at 360 °C with peak temperature of 427.7 °C and weight loss of about 79.78%. They made unusual observation when the pyrolysis of biomass–PET mixtures occurred at temperatures higher than pure biomass and PET. Moreover, they found the range of activation energy values of PET was higher than the biomass used. This demonstrated that the decomposition of biomass started at lower temperatures. They attributed this difference to the structural differences between them. For pure PET, they obtained a reaction in two stages. The first stage had temperature range of 373–443 °C, conversion range of 0.4–81%, and activation energy value of 347.4 kJ/mol, while the second stage had a temperature range of 448–503 °C, conversion range of 88–99%, and activation energy value of 172.6 kJ/mol. They presented two tables of activation energy values for different pure biomasses (CS, HS, SFR, and ER), PET, and the biomass–PET mixture. The average activation energy value of CS, PET, and CS–PET decompositions were 71, 260, and 95.7 kJ/mol, respectively.

Lee et al. (2018) [13] studied the catalytic decomposition of mixed torrefied cork oak (TOak) and HDPE using three different catalysts (mesoporous HY1, mesoporous HY2, and microporous HY). They stated that the catalytic pyrolysis of HDPE over HY had two reactions within the range of 200–300 and 300–400 °C with peak temperature of 392 °C, which was lower than the pure FDPE temperature of 477 °C. This behavior was different from the catalytic copyrolysis of Oak/HDPE and TOak/HDPE over HY. In these cases, the peak temperature was shifted higher to 411 and 414 °C, respectively. Diaz Silvarrey and Phan (2016) [14] found that the decomposition of PET by TGA started earlier than most other polymers with a thermal decomposition range of 395–520 °C. They used the KAS method to obtain 197.61 kJ/mol as the average value of activation energy and 4.84 × 10^14^ s^−1^ as the pre-exponential factor.

Xiang et al. (2018) [15] investigated the catalytic decomposition of cobalt-modified ZSM-5 catalyst on copyrolysis of lignocellulosic biomass (rice straw (RS)) and linear low-density polyethylene (LLDPE). They stated that there was only one peak for the pyrolysis of pure RS at 341.39 °C and pure LLDPE at 491.53 °C, two peaks for the copyrolysis of RS–LLDPE at 339.57 and 470.06 °C, and two peaks for the pyrolysis of RS–LLDPE with Co/ZSM-5 catalyst at 336.32 and 451.73 °C. They calculated the activation energy of the pyrolysis of LLDPE, RS, RS–LLDPE, and catalytic RS–LLDPE as 212.23, 79.61, 34.27, and 41.04 kJ/mol, respectively. Das and Tiwari (2019) [5] used PET collected from soft drink bottles to evaluate the degradation kinetics of nonisothermal TGA runs with five different heating rates. They obtained activation energy values in the range of 203–355 kJ/mol using the Vyazovkin method and established the F1 reaction model using Criados’ master plot.

Osman et al. (2020) [16] employed TGA analysis for PET pyrolysis with five heating rates with a ratio of 16 between the lowest (0.5 K/min) and highest (8 K/min) rates. They applied different isoconversional methods (integral Friedman, ASTM-E698, and integral FWO) and concluded that the activation energy was between 165 and 195 kJ/mol.

Ali et al. (2020) [17] studied the catalytic cracking kinetics of polystyrene with synthesized copper oxide as a catalyst. They used Friedman, KAS, FWO, and Coats–Redfern to obtain the kinetic parameter values. They concluded that the low value of the activation energy would help to run experiments at lower temperatures. Lai et al. (2021) [18] examined the performance of coal ash (fly and bottom ash) with different blending ratios for the catalytic pyrolysis of LLDPE. They showed that 15 wt% bottom ash had higher effect on catalytic activity compared to 15 wt% fly ash. Yap et al. (2022) [19] used two different catalysts (HZSM-5 zeolite and natural limestone (LS)) for copyrolysis of HDPE and *Chlorella vulgaris* mixture. They concluded that the combined HZSM-5/LS catalyst had the best effect in the copyrolysis of the HDPE and *Chlorella vulgaris* mixture, with the activation energy decreasing from 144.93–225.84 to 75.37–76.90 kJ/mol. Kokuryo et al. (2022) [20] studied the catalytic degradation of LDPE using two types of catalysts (zeolite β and zeolite MTES-β) at 5 K/min heating rate. They concluded that zeolite MTES-β pushed LDPE to decompose at a lower temperature than zeolite β. Dourari et al. (2022) [21] and Tarchoun et al. (2022) [22] developed new composites with different materials and tested them using various analytical techniques. They concluded that there was a big drop in the activation energy of the thermal behavior.

In this study, the activation energy of PET obtained by catalytic thermal decomposition was calculated by four model-free methods and two model-fitting techniques. In order to study the effect of the zeolite β catalyst, these kinetic parameter values were compared with the pure PET pyrolysis values collected from the literature survey. Furthermore, the catalytic pyrolysis behavior of PET was determined for the first time by a strongly promising ANN model.

## 2. Materials and Methods

### 2.1. Materials

PET polymer samples were collected from Recycled Plastic (Ipoh, Malaysia). The two main analyses of the polymer (shown in Table 1) were performed using PerkinElmer thermal analyzer and PerkinElmer elemental analyzer (Waltham, MA, USA). Zeolite β (its main properties are shown in Table 2) catalyst in powder form was obtained from Alfa Aesar (Ward Hill, MA, USA).

All the details related to preparing the powders and sample weighing can be found in [23].

### 2.2. Catalytic Thermal Decomposition of PET

The 1020 series TGA7 thermogravimetric analysis was used to meet the objective of this research. Under the control of the system controller, the changes in weight due to chemical reaction were measured. PET polymer pellets were ground into powders before being fed to TGA. A good reproducibility was achieved in the range of 5–100 mg. Therefore, samples weighing 10 mg were used throughout the study. The samples were heated at 2, 5, 10, and 20 K/min under nitrogen at 20 mL/min within the temperature range of 298–873 K at constant composition of zeolite β to PET (20/80 wt/wt).

### 2.3. Kinetic Theory

The rate of reactions (*r*) can be expressed as a product of a temperature (*T*)-dependent function, *K*(*T*), and a composition- or conversion (*α*)-dependent term, *f*(*α*), as follows [24]:(1)$$r=\frac{d\alpha }{dt}=K\left(T\right)f\left(\alpha \right)$$
(2)$$\alpha =\frac{w_{o}-w}{w_{o}-w_{f}}\;$$

The definitions of *α*, *t*, *T*, *w_o_*, *w_o_*, and *w_f_* can be found in [23]. The temperature-dependent term, which is the reaction rate constant (*K*), is assumed to obey the Arrhenius relationship:(3)$$K\left(T\right)=A_{0}\exp\left(-\frac{E}{RT}\;\right)$$
where *E* is the activation energy (kJ/mol), *A*_0_ is the pre-exponential factor *K*^−1^, and *R* is the universal gas constant (8.314 J/mol.K). If it is assumed that a simple *n*th-order kinetic relationship holds for the conversion-dependent term, then the following applies:(4)$$f\left(\alpha \right)={\left(1-\alpha \right)}^{n}\;$$

Substituting Equations (3) and (4) in (1) yields the following:(5)$$r=\frac{d\alpha }{dt}=A_{0}\exp\left(-\frac{E}{RT}\;\right){\left(1-\alpha \right)}^{n}$$

All the published studies have used either single or multiple thermograms at different heating rates with either differential or integral methods for calculating kinetic parameters from TGA data [24]. However, in this work, two different sets of methods were examined and compared with the published available data. One set used multiple thermograms, called model-free methods, which included Friedman, FWO, KAS, and Starnik (different heating rates and constant conversion). The second set used single thermograms, called model-fitting methods, which included Coat–Redfern and the Criado equation. Derivations of all these equations with the common solid-state thermal reaction mechanisms *g*(*α*) used in the Coats–Redfern method can be found in [23,25,26].

### 2.4. Topology of ANNs

There is a common problem in modeling any engineering process, especially if the final model is very complicated with high nonlinearity and too many parameters. In these cases, ANN can be considered as another alternative option. Muravyev et al. (2021) [27] reviewed the main concept of ANN for application in pyrolysis, thermal analysis, and kinetic studies. They highlighted that there is no single mathematical technique to resolve all pyrolysis problems.

An ANN architecture is ordered in three consecutive layers of neurons: input, hidden, and output. In each layer, there is a weight matrix, bias vector, and output vector [28]. The variables influencing the structure of the network should be fixed and the whole problem area should be represented outside the ANN. ANN architecture should have a number of layers, with each layer having a transfer function and connections between the layers. The type of problem presented will determine the best architecture. The number of neurons in the hidden layer, the number of hidden layers, the momentum rate, and the learning rate are part of the optimizing parameters.

In this research, the performance of an ANN model was monitored by the following statistical parameters: average correlation factor (R^2^), root mean square error (RMSE), mean absolute error (MAE), mean bias error (MBE), and correlation coefficient (R) statistical correlations [1,29,30,31].

The remaining mass percentage of PET was measured by the ANN model. There are some advantages and maybe some disadvantages in using ANN. Some of these advantages include the ability to work with linear and nonlinear relationships and learn these relationships directly from the data used, while some of these disadvantages including the need for large memory and computational efforts for fitting [32].

## 3. Results and Discussion

### 3.1. TG–DTG Analysis of PET

TG and DTG at different heating rates of the catalytic pyrolysis of PET by zeolite β are shown in Figure 1. The thermograms were similar for all different heating rates. Osman et al. (2022) [33] and Farrell et al. (2021) [34] emphasized that TGA experiments should be run according to the International Confederation for Thermal Analysis and Calorimetry (ICTAC) instructions in order to collect the proper data for calculation of kinetic parameters. Therefore, they used 16 as the ratio between the highest and lowest heating rates. Muravyev and Vyazovkin (2022) [35] reviewed the most proper options that should be chosen as advised by the International Confederation for Thermal Analysis and Calorimetry (ICTAC). ICTAC mainly recommends using model-free methods, where the data should be collected at more than three different heating rates. ICTAC emphasized that TGA experiments should be run according to its instructions in order to collect the proper data for kinetic parameter calculations. Because the TG curve showed only one peak reaction, single-step thermal decomposition as recommended by ICTAC was considered to derive the proper kinetic parameters [36].

Catalytic thermal decomposition onset, peak, and final temperatures, which were obtained from the two curves, moved to higher temperature when the heating rate increased from 2 to 20 K/min. Diaz Silvarrey and Phan (2016) [14] mentioned that as the heating rate increases, the peak of DTG will be bigger.

It has been shown that all plastic polymers except PVC degrade by only one reaction, including PET [5,6]. As shown in Table 3, the onset, peak, and final temperatures of catalytic cracking of PET by zeolite β catalyst were smaller than those of cracking decomposition of pure PET. For example, the onset, peak, and final temperatures of catalytic cracking at 10 K/min were 625, 710, 740 K in this study, while they were 671, 711, 748 K in [5] and 643, 714, 775 K in [6]. Al-Salem et al. (2017) [4] highlighted that catalytic cracking is more promising than thermal degradation as it requires lower energy.

### 3.2. Model-Free Kinetics Calculation

Equation (5) is considered as the starting equation from which all the other equations can be attained. Four different types of isoconversional methods were implemented to calculate the main kinetic parameter (activation energy (*E_a_*) in kJ/mol). The derivation of these four methods can be found in [23]. Friedman, FWO, KAS and Starnik at conversions ranging from 0.1 to 0.8 and the activation energies values for the four methods are presented in Figure 2 and Table 4. From these, it can be seen that the values of activation energy with the Friedman’s method (average 167.5 kJ/mol) were slightly higher than and totally different from the rest three, where the values were homogeneous with an average of 140.2 kJ/mol. These two values were still much less than what has been published for pure PET pyrolysis. *E_a_* values in the range of 203–355 [5], 165–195 [37], and 185–230 [14] kJ/mol have previously been reported for pyrolysis of PET. Therefore, the lowest activation for PET degradation in the presence of zeolite β catalyst proves the suitability of using a catalyst at lower temperatures. This finding has been supported by many researchers [4,9,11,17,20]. Al-Salem et al. (2017) [4] reviewed and confirmed that a catalyst lowers the required pyrolysis temperature for all polymers. As shown in Table 3, there was no trend in the activation energy, except that the Freidman methods started at lower values (130 kJ/mol at α = 0.1), increased over the period of degradation (188 kJ/mol at α = 0.6), and decreasing until the end (136 kJ/mol at α = 0.8). For FWO, KAS, and Starnik, the trend was the same, with a steady increase from an average of 93.33 kJ/mol (α = 0.1) to an average of 164 kJ/mol (α = 0.7) and then a decrease to 156 kJ/mol (α = 0.8).

Generally, these methods are considered more reliable compared to model-fitting methods as they obtain consistent activation energy values for the nonisothermal data [37,38]. They also help to show the complexity of multireaction as the calculation of the activation energy depends on the conversion [39]. Moreover, Özsin and Pütün (2019) [40] found that the activation energy values for the four different methods depended heavily on the conversion value. From the calculations and graphs, it became clear that there is a need in the future to conduct additional experiments with more heating rates to improve some low values of R^2^ in Table 4 caused by the ratio of 10 between the highest and lowest heating rates.

### 3.3. Model-Fitting Kinetics Calculation

Two model-fitting (Criado and Coats–Redfern) methods were implemented to study the kinetics of catalytic cracking of PET. For more details relating to derivation, please refer to [26]. The first model-fitting method (Coats–Redfern) was tested by plotting $\ln\left[\frac{g\left(\alpha \right)}{T^{2}}\right]$ against 1/T. The most suitable formula of function g(α) common solid-state reaction mechanism was selected and presented in [41,42]. Appendix A lists the kinetic parameter (*E_a_ kJ*/*mol* and *lnA*) and the average correlation factor (*R*^2^). It can be seen from the Appendix A that there were big differences in the *E_a_* between these 16 different reaction mechanisms. In the next stage, these data results were connected to the second model-fitting (Criado method). The Criado method was utilized to calculate the reaction model. In the Criado method, the left-hand side of the Criado equation is called a reduced theoretical curve, and the right-hand side represents the experimental data. To use this method, comparison between these two sides were carried out to help us define the exact kinetic model describing the reaction. Appendix A shows the curves obtained from the Criado method at different heating rates only for the reaction mechanisms close to the experimental curve. Appendix A and Table 5 reveal that the experimental data matched first-order reaction (F1) or two dimensional phase interfacial reaction (R2) for 2, 5, and 10 K/min heating rates and (F1) or (R2) or two dimensional nucleation and growth reaction (A2) for 20 K/min. Table 5 reveals that there was a range of activation energy values between 104 to 242 kJ/mol with an average of 157 kJ/mol depending on different reaction mechanisms. Das and Tiwari (2019) [5] showed that PET degradation under nitrogen follows many reaction models (A2, A3, A4, and F1) for all the heating rates.

### 3.4. Catalytic Pyrolysis Prediction by ANN Model

ANN model was developed to monitor the catalytic pyrolysis of PET. Feedforward with backpropagation was developed to predict the remaining weight fraction based on 983 experimental datasets with heating rate and temperature as the input variables.

The collected data is usually divided into three subsets: training set, validation set, and finally test set [43]. Each set has its own role in the whole process.

The whole data comprising 983 sets were randomly divided into three sets as follows: 70% used for training, 15% used for validation, and 15% used for testing. More details about the data are shown in Table 6. Osman and Aggour (2002) [44] mentioned that collecting large sets of data can lead to a model with high accuracy compared to the original data. To obtain the best performance of ANN structure, it is necessary to fix: (1) the number of hidden layers, (2) transfer function for output and hidden layer/s, and (3) number of neurons in each hidden layer. A comparison was then carried out between different ANN structures with variable number according to the above points.

The value of R was used as the main criterion for judging the most efficient structure to estimate the remaining weight fraction as the output variable for 23 runs. A total of 13 runs out of the 23 runs resulted in high value of R. These 13 runs were moved to the second stage comparison, where 12 new simulation sets were tested again against the value of R. Table 7 shows the 12 new sets that were collected for the four heating rates.

The final and best ANN structure (2-15-15-1), as shown in Figure 3, was selected after the two comparison steps. It had 15 neurons for each two hidden layers (tangsig–tangsig transfer functions) with linear transfer function for the output layer. The efficiency and the accuracy of the ANN output is heavily dependent on the number of neurons in the hidden layers. To avoid underfitting and the overfitting, one must select the number of neurons in such a way that the performance function will eventually obtain the optimum value [43,45]. Al-Wahaibi and Mjalli (2014) [38] concluded that the number of neurons in the hidden layer is a very crucial parameter in predicting the accuracy of the ANN output, and using very few numbers may result in something called underfitting. Qinghua et al. (2018) [40] also pointed out that the number of neurons in the hidden layer may cause training error if the number is too small or overfitting problem if the number is too high. The Levenberg–Marquardt algorithm is commonly suited for this type of network [46]. The ANN model was run with different algorithms, namely, scaled conjugate gradient and Bayesian regularization, to check that the test was relevant. As shown in Figure 4, all the comparative results were close to the diagonal, which proved good agreement between the ANN values (Y-axis) and real values (X-axis).

The performance of the current (2-15-15-1) model in predicting remaining weight fraction was examined by calculating the four statistical parameters. As shown in Table 8, all statistical measurements of deviations were notably low for the best ANN model. This meant the selected ANN model could efficiently predict the output variable within an acceptable limit of error. After training the ANN with 983 sets, the recommended architecture (2-15-15-1) was simulated with 12 new input datasets as delineated in Table 7. In this stage, new input data were entered, and the network produced the new simulated output. Figure 5 compares the simulated ANN with the measured output and Table 9 lists the R^2^, RMSE, MAE, and MBE values. As can be seen, R^2^ was very high (>0.9999), while RMSE, MAE, and MBE were very low. Figure 6 presents the error histogram generated for the data. As can be seen, the error was again only in a narrow range (−0.00091 to 0.000402).

## 4. Conclusions

As expected, TG and DTG showed the same shapes and trends for different heating rates, with only one main reaction. To obtain the kinetic parameters, six methods, namely, four model-free (Friedman, FWO, KAS, and Starnik) and two model-fitting (Coats–Redfern and Criado) methods were used as the main approach. A highly efficient ANN model was developed as a second approach to predict the catalytic decomposition of PET.

The two approaches were implemented to model the TGA kinetics data. In the first approach, the six different methods were used to approximate the TGA data with a straight line. Values of the activation energies obtained by Friedman, FWO, KAS, and Starnik methods were 167.5, 142.75, 138.75, and 139.13 kJ/mol, respectively. The average calculated values of activation energies by Coats–Redfern and Criado methods were in the range of 104–242 kJ/mol. For future work, more heating rates are needed to fill the gap in the results. Natural polymers, such as cellulose, are also recommended to enhance the catalytic cracking of polymer plastic waste.

An ANN with 2-15-15-1 structure and tangsig–tangsig hidden layers was determined as the most promising network as it could predict the remaining weight fraction very precisely with high regression coefficient value. The recommended network was reused with new input and still showed results that were close to the experimental values with high R and very low RMSE, MAE, and MBE.

## Figures and Tables

**Figure 1 polymers-15-00070-f001:**
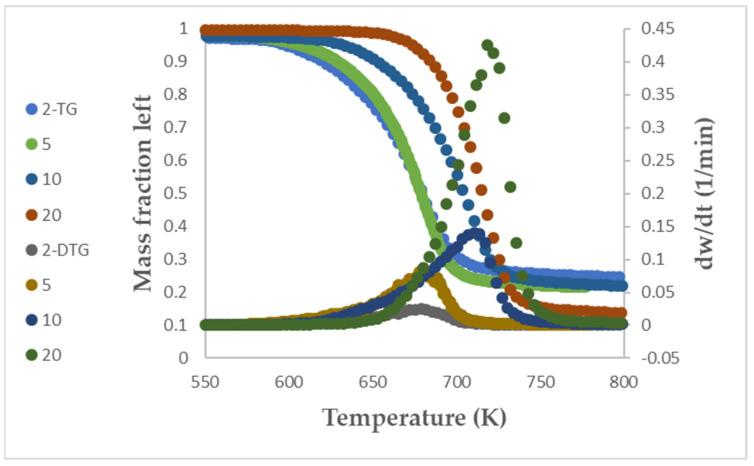
Thermogravimetric analysis (TGA) and derivative thermogravimetric (DTG) curves for catalytic cracking of PET at different heating rates.

**Figure 2 polymers-15-00070-f002:**
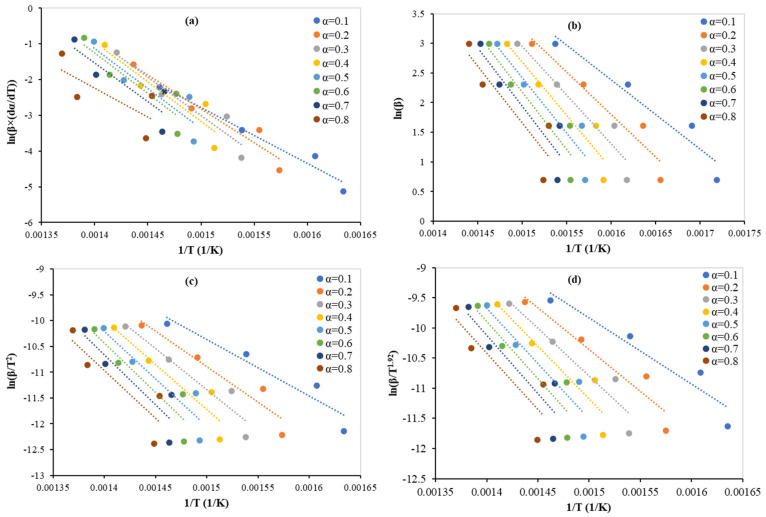
Linear regression lines of catalytic cracking of PET at different conversions by (**a**) Freidman, (**b**) FWO, (**c**) KAS, and (**d**) Starnik methods.

**Figure 3 polymers-15-00070-f003:**
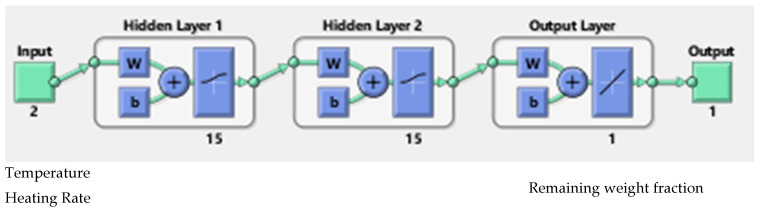
Topology of the best selected (2-15-15-1) network.

**Figure 4 polymers-15-00070-f004:**
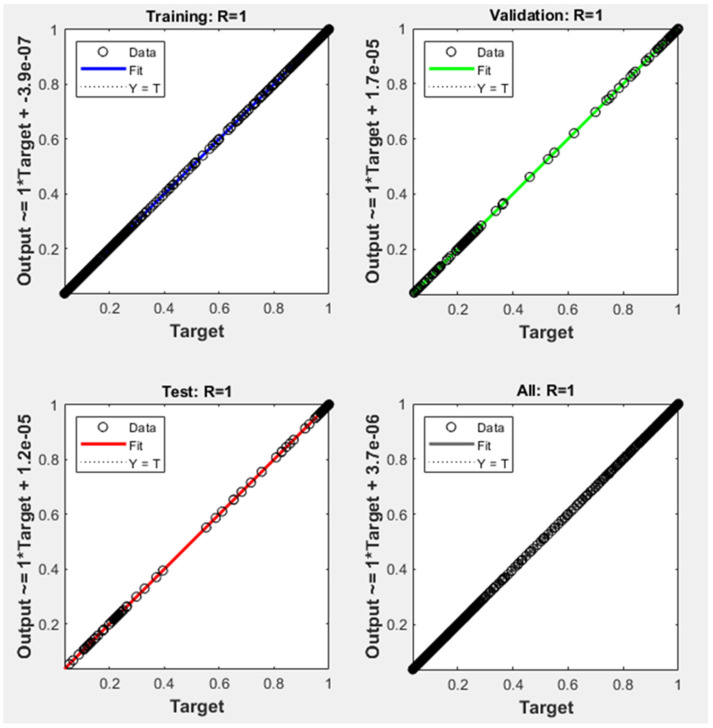
Regression of training, validation, and test plots for the selected (2-15-15-1) model.

**Figure 5 polymers-15-00070-f005:**
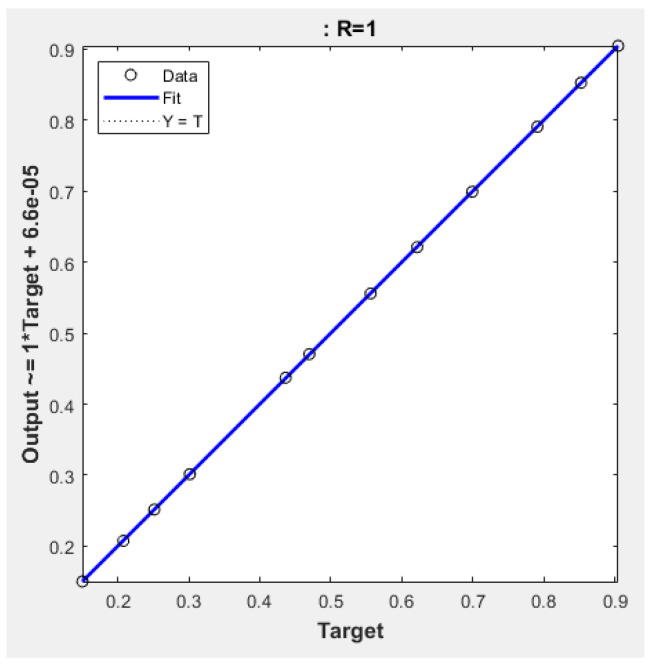
Regression of simulated data for the selected (2-15-15-1) model.

**Figure 6 polymers-15-00070-f006:**
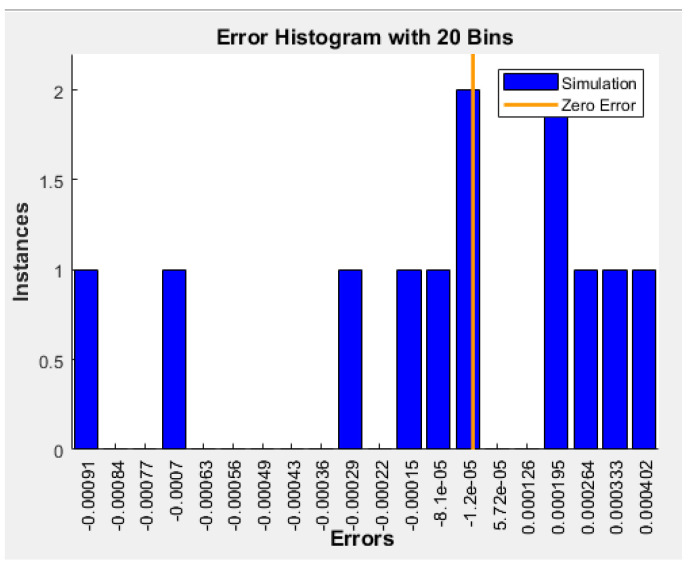
Error histogram of simulation data.

**Table 1 polymers-15-00070-t001:** The two main analyses of PET.

Plastic	Proximate Analysis, wt%			Ultimate Analysis, wt%			
	Moisture	Volatile	Ash	C	H	N	O
PET	0.523	88.231	11.246	64.256	4.367	0	31.377

**Table 2 polymers-15-00070-t002:** The main properties of Zeolite β.

S.A. m^2^/g	SiO_2_:Al_2_O_3_	Pore Volume 1.7–300 nm, m^2^/g
680	25:1	0.127

**Table 3 polymers-15-00070-t003:** Comparison between catalytic cracking characteristics of PET at different heating rates (this work) and pyrolysis of pure PET [5,6,16].

Heating Rate K/min	Main Reaction (Catalytic Cracking)				Pyrolysis of Pure PET				References
	Onset (K)	Peak (K)	Final (K)	Mass Loss (%)	Onset (K)	Peak (K)	Final (K)	Mass Loss (%)	
2	575	675	700	70	623	667	694	100	Osman et al. (2020) [16]
5	600	685	710	75	658	700	723	80	Das and Tiwari (2019) [5]
10	625	710	740	78	671 (643)	711 (714)	748 (775)	80	Das and Tiwari (2019) [5], Yang et al. (2001) [6]
20	650	720	750	85	681	721	759	80	Das and Tiwari (2019) [5]

**Table 4 polymers-15-00070-t004:** Activation energies by Friedman, FWO, KAS, and Starnik methods.

Conversion	Friedman		FWO		KAS		Starnik	
	E (kJ/mol)	R^2^	E (kJ/mol)	R^2^	E (kJ/mol)	R^2^	E (kJ/mol)	R^2^
0.1	130	0.9621	97	0.9286	91	0.9116	92	0.9124
0.2	158	0.9271	119	0.9278	115	0.9142	115	0.9148
0.3	178	0.9065	137	0.9184	132	0.9052	133	0.9058
0.4	185	0.8493	149	0.898	145	0.8833	145	0.844
0.5	186	0.8041	156	0.8741	153	0.8573	153	0.8581
0.6	188	0.8034	162	0.8532	158	0.8347	159	0.8355
0.7	179	0.7574	164	0.8298	161	0.8092	161	0.8101
0.8	136	0.5434	158	0.7905	155	0.7654	155	0.7664
Average	167.5	0.8192	142.75	0.8775	138.75	0.8601	139.13	0.8559

**Table 5 polymers-15-00070-t005:** Activation energy of PET catalytic pyrolysis at different heating rates by Coats–Redfern method.

Heating Rate (K/min)	E (kJ/mol)	Mechanism
2	146	F1 (first-order chemical reaction)
	104	R2 (two-dimensional phase interfacial reaction)
5	179	F1 (first-order chemical reaction)
	141	R2 (two-dimensional phase interfacial reaction)
10	139	F1 (first-order chemical reaction)
	123	R2 (two-dimensional phase interfacial reaction)
20	242	F1 (first-order chemical reaction)
	115	A2 (two-dimensional nucleation and growth reaction)
	224	R2 (two-dimensional phase interfacial reaction)

**Table 6 polymers-15-00070-t006:** Distribution of 983 datasets (first step: training set) and 12 dataset (second step: simulation set) for heating rates of 2, 5, 10, 20 K/min.

Heating Rate (K/min)	Training Set No.			Simulation set No.
	Training	Validation	Test	
2	248			3
5	248			3
10	246			3
20	241			3
Total	983			12

**Table 7 polymers-15-00070-t007:** ANN-predicted remaining weight values of the new (simulation step) input.

No.	Input Data		Output Data
	Heating Rate (K/min)	Temperature (K)	Remaining Weight (Fraction)
1	2	702.1	0.30153
2	2	682.2	0.46976
3	2	618.9	0.90445
4	5	704.8	0.25160
5	5	670.4	0.62158
6	5	641.3	0.85232
7	10	968.5	0.15028
8	10	687.2	0.69914
9	10	700.4	0.55610
10	20	718.6	0.43629
11	20	697.8	0.79087
12	20	732.4	0.20794

**Table 8 polymers-15-00070-t008:** Statistical parameters of the best (2-15-15-1) model.

Set	Statistical Parameters			
	R^2^	RMSE	MAE	MBE
Training	1.000	1.820 × 10^−4^	9.800 × 10^−5^	−4.505079 × 10^−7^
Validation	1.000	3.680 × 10^−4^	1.680 × 10^−4^	−0.0000295
Test	1.000	2.580 × 10^−4^	1.390 × 10^−4^	8.015 × 10^−6^
All	1.000	2.310 × 10^−4^	1.140 × 10^−4^	−0.00000353

**Table 9 polymers-15-00070-t009:** Statistical parameters of the (2-10-10-1) model for the simulated data.

Set	Statistical Parameters			
	R^2^	RMSE	MAE	MBE
simulated	1.000	3.903 × 10^−4^	2.843 × 10^−4^	6.360 × 10^−5^

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
