# Peer review of "Catalytic Pyrolysis of PET Polymer Using Nonisothermal Thermogravimetric Analysis Data: Kinetics and Artificial Neural Networks Studies"

_polymers, 2022, doi:10.3390/polym15010070_

Round 1

Reviewer 1 Report

Reference paper: polymers-2115900

The authors reported a detailed kinetic analysis of the catalytic pyrolysis Polyethylene Terephthalate (PET) polymer using isoconversional methods and artificial neural network. The results have high reference value for related research and application. Therefore, I recommend accepting it after major revision.

Comments:

1-      The general structure is acceptable, the results seem good as well and the discussion is fine. However, language should be thoroughly revised as some of the sentences are confusing and some errors can be found.

2-        The importance of the investigation should be justified and highlighted in the abstract and the introduction parts.

3-        The applicability of the present study should be addressed.

4-        Some relevant information about the used model-free and model-fitting kinetic methods should be added in the introduction part. The choice of these kinetic approaches should be also justified.

5-        The used mass fraction of Zeolite catalyst (20%) should be justified. The preparation pathway of PET/catalyst should be also reported (a schematic illustration of the preparation procedure will be helpful).

6-        The originality and the novelty of the paper needs to be further clarified in the introduction part.

7-        Regarding the performed kinetic calculations, have the authors followed the reference the recent ICTAC kinetics committee recommendations. I recommend the authors to discuss the ICTAC Kinetic committee recommendations on thermal decomposition kinetics (https://doi.org/10.1016/j.tca.2022.179384).

8-        In the most general way, model-fitting methods provide less accurate kinetic results. Could the authors comment this statement?

9-        I think that the obtained linear correlation coefficient (R2) is low, and we can not confirm the accuracy of the obtained activation energy.

10-    I see that the authors did not calculate the preexponential factor. Why ?

11-    The errors associated with the determined Arrhenius parameters (activation energy and preexponential factor) by all considered approaches (model-free, model fitting and compensation effect) should be calculated and presented in the manuscript. (you can see and cite the following papers 1- https://doi.org/10.3390/catal12121573; 2- https://doi.org/10.3390/molecules27206945)

12-    The references part should be updated and homogenized again according to the journal style.

Reviewer 2 Report

This manuscript reports the kinetic study of the catalytic pyrolysis of zeolite and polyethylene terephthalate (PET) mixture. The authors use a couple of kinetic methods, including the neural network analysis. Although the results are of interest to the community and should be delivered, the paper needs a serious revision to improve its discussion.

abstract: "153.85 kJ/mol" - please provide less digits, given the complexity of the studied process

introduction: some text is underlined, please, fix it at revision

intro: "TGA is one of the thermal analysis methods and most popular for calculating the kinetic parameters of plastics pyrolysis." - disabbreviate it at first appearance. Additionally, this and other general statements can be supplemented by the recent overview on the kinetic measurements for pyrolysis studies (https://doi.org/10.3390/thermo2040029)

intro: "In this literature, it will be focused only for the topic of “kinetic of PET by TGA data” and “kinetics of catalytic cracking of different plastic wastes polymers by TGA data”." - not sure, what these phrase mean, the words within quotes are the search queries?(where? specify it)

intro: ". Diaz Silvarrey and Phan (2016) [14]" - it looks like hyperlink, fix it

intro: please give the proper background by discussing the emerging trend on using of neural networks for thermal and kinetic analysis (i.e., the recent review https://doi.org/10.3390/molecules26123727)

experimental: Tables 1,2 (and other Tables later in text) are refered as a hyperlinks in text, that should be corrected

results: Figure 1 - the figure axes should be according to MDPI style

Round 2

Reviewer 1 Report

The authors corrected all the necessary issues. I believe that the manuscript can be published in its current form.

Reviewer 2 Report

The authors address all my comments in the revised version. Paper can be published as is